# AERO: Enhancing Sharding Blockchain via Deep Reinforcement Learning for Account Migration

## Abstract

Sharding blockchain networks face significant scalability challenges due to high frequencies of cross-shard transactions and uneven workload distributions among shards. To address these scalability issues, account migration offers a promising solution. However, existing migration solutions struggle with the high computational overhead and insufficient capture of complex transaction patterns. We propose AERO, a deep reinforcement learning framework to facilitate efficient account migration in sharding blockchains. AERO employs a prefix-based grouping strategy to enable group-level migration decisions and capture complex transaction patterns and relationships between accounts. We also implement a sharding blockchain system called AEROChain, which integrates AERO and aligns with the blockchain decentralization principle. Extensive evaluation with real Ethereum transaction data demonstrates that AERO improves the system throughput by 31.77% compared to existing solutions, effectively reducing cross-shard transactions and balancing shard workloads.

## CCS Concepts

• **Theory of computation → Algorithmic mechanism design**; • **Computing methodologies → Reinforcement learning**; • **Computer systems organization → Peer-to-peer architectures**.

## Keywords

Blockchain, Sharding, Account migration, Reinforcement learning

### ACM Reference Format:

Anonymous authors. 2025. AERO: Enhancing Sharding Blockchain via Deep Reinforcement Learning for Account Migration. In *Proceedings of the ACM Web Conference 2025 (WWW '25)*. ACM, New York, NY, USA, 11 pages. https://doi.org/XXXXXXX.XXXXXXX

## 1 Introduction

Blockchain technology has rapidly evolved as a cornerstone of the emerging Web 3.0 [30]. By maintaining immutable transaction records and ensuring trustless interactions, blockchain is vital for creating a transparent and secure decentralized web [12]. However, despite its revolutionary, blockchain faces significant scalability challenges [36]. Conventional blockchain networks are limited in transaction processing capacity [10]. As they grow with more users

and applications, scalability bottlenecks hinder their widespread adoption and the full potential of Web 3.0 [21].

To address the blockchain scalability issue, sharding has been proposed as a promising solution [20]. Sharding partitions the blockchain network into multiple smaller, manageable segments called shards. Each shard simultaneously processes a subset of blockchain transactions and smart contracts, while periodically reassigning and maintaining shard nodes to ensure security. This parallel processing approach theoretically increases the network's overall capacity proportionally to the number of shards, thereby enhancing blockchain scalability.

Nevertheless, sharding introduces its own series of challenges [16]. One of the primary issues is the high frequency of Cross-Shard transactions (CSTXs), which occur when transaction accounts are located on different shards [10, 18]. Processing CSTXs is more time-consuming and resource-intensive than intra-shard transactions, because it requires coordination between shards and can lead to increased latency [32]. Moreover, the uneven workload distribution across shards is also a significant concern [17]. Due to the power-law distribution often observed in transactions [14], some shards may become overloaded while others remain underutilized, leading to inefficiencies and potential bottlenecks within the network.

Substantial research focuses on account migration mechanisms to mitigate the challenges of CSTXs and workload imbalance in sharding blockchain systems [9, 10, 16, 17]. The account migration involves periodically redistributing user accounts across shards to reduce CSTXs and balance the workload. Some work utilizes graph partitioning and account segmentation strategy to optimize the assignment of accounts [10]. However, the computational overhead of graph partitioning algorithms leads to performance degradation, and this work introduces significant complexity in maintaining sub-accounts by account segmentation.

Motivated by the need for an efficient and decentralized account migration mechanism, we explore applying deep reinforcement learning (DRL) [27] to this problem. DRL is highly effective in handling sequential decision-making tasks and has demonstrated significant potential in optimizing complex systems with expansive state and action spaces [24]. In the context of account migration, the account migration sequence can be treated as a decision-making process where the objective is to assign accounts to shards in a manner that minimizes CSTXs and balances the workload. Existing DRL-based sharding solutions, such as SPRING[18], process one account at a time, which leads to an enormous action space due to the vast number of accounts and shards. This approach can be inefficient and may not scale well with larger blockchain networks. However, without careful design, the DRL agent is likely to struggle with the vast action space due to the large number of account addresses, which in turn limits its effectiveness.

Based on the above analysis, we propose **AERO**, a novel DRL framework for efficient account migration in sharding blockchain

networks. AERO introduces a prefix-based granularity approach, grouping accounts based on common prefixes to not only reduce the action space significantly but also enable efficient migration of a large number of accounts. With this approach, AERO can make group-level migration decisions, rather than handling each account individually. To demonstrate the feasibility and effectiveness of integrating AERO into a sharding blockchain system, we design and develop **AEROChain**. AEROChain ensures the decentralization of AERO's operation through its consensus mechanism. Extensive experiments using real Ethereum transaction data are conducted to evaluate the performance of AERO. The results indicate that AERO significantly reduces the number of CSTXs and achieves a more balanced workload distribution among shards compared to existing state-of-the-art algorithms. Specifically, AERO improves system throughput by 31.77% compared to other strategies, showing its effectiveness in enhancing blockchain system overall performance.

In summary, our contributions are as follows:

- We propose **AERO**, a DRL framework designed to efficiently generate account migration plans by reducing the action space through a prefix-based granularity approach. AERO aims to reduce CSTXs and achieve balanced workloads across shards in the sharding blockchain system.
- We implement a sharding blockchain called **AEROChain**, integrating AERO and detailing the complete workflow to demonstrate the feasibility and adherence of the framework to blockchain decentralization principles.
- We perform extensive experiments using real transaction data, showing that AERO outperforms existing solutions by improving throughput by 31.77%, reducing CSTXs and improving workload balance.

## 2 Background and Related Work

### 2.1 Sharding Blockchain with Deep Reinforcement Learning Approaches

Sharding technology has become a crucial solution for improving blockchain scalability by partitioning the blockchain network into smaller shards that process transactions in parallel [22]. With the adoption of Practical Byzantine Fault Tolerance (PBFT) for intra-shard consensus, the performance of sharding blockchains has steadily improved, achieving near-linear throughput scalability as the network grows [5, 18, 23, 26]. In recent years, sharding has already become a core component of the mainstream blockchain to enhance blockchain scalability and throughput by splitting the blockchain network into multiple interconnected shards [28].

Deep reinforcement learning (DRL) [1] integrates reinforcement learning with deep learning to address complex sequential decision-making problems. DRL operates within the framework of a Markov Decision Process (MDP), defined by a 4-element tuple: a set of states, actions, transition probabilities, and rewards. The agent interacts with the environment by selecting actions, transitioning between states, and receiving feedback in the form of rewards. The primary objective of reinforcement learning is to train a policy that maximizes cumulative rewards over time. Through iterative interactions, the agent learns to refine its policy, improving its decision-making by either exploring the environment or exploiting past knowledge. DRL further enhances this process by utilizing

deep neural networks to model the policy, enabling the agent to identify complex patterns and relationships in the environment, making it well-suited for dynamic and intricate tasks.

DRL has already found applications in the sharding blockchain network, addressing challenges such as address placement, resource allocation, and transaction processing [18, 19, 31, 34]. SPRING[18] applies DRL to improve address placement strategies, enabling more efficient transaction processing by learning policies over time. SkyChain[34] utilizes DRL to optimize resource allocation, aiming to enhance transaction throughput and reduce latency. Additionally, Lin[19] introduces DRL to enhance dynamic shard formation and improve communication efficiency in the federated learning context. However, these approaches do not specifically address account migration, which is crucial for minimizing CSTXs and balancing workloads among shards.

### 2.2 Account Migration

Account migration protocols in sharding blockchains are essential for maintaining scalability by redistributing account states across different shards [22]. Early approaches rely on constructing and analyzing transaction graphs [2, 15]. Among these related works, transactions are represented as edges connecting account nodes, and the corresponding transaction graphs are partitioned using graph partitioning or community detection methods to determine optimal shard allocations for accounts. However, maintaining and processing transaction graphs on a blockchain presents significant challenges. The vast number of transactions generates enormous graphs, requiring substantial storage and computational resources. Notably, previous work [16] utilizes a community-aware account partition algorithm to balance the shard workload and reduce CSTX ratios but struggles to maintain the trade-off between each other. BrokerChain [10] offers a broker-based account migration approach to reduce CSTXs. Nonetheless, BrokerChain's reliance on a broker network raises concerns about centralization and bottlenecks, as the system becomes dependent on the availability of these brokers.

Apart from graph partitioning algorithms, several advanced techniques have been applied to the account migration problem. Monoxide [29] employs asynchronous consensus zones to scale out blockchains, improving throughput and capacity. Another method introduces locking schemes to prevent double-spending and race conditions during the migration [9]. Although these approaches can ensure security during the transfer process, their reliance on locks increases complexity, reducing system throughput. Moreover, they do not fundamentally resolve the issues of uneven shard load and high CSTX volumes. LB-Chain [17] introduces a load-balancing mechanism that uses LSTM network predictions [8] to distribute accounts across shards. While it effectively reduces the uneven shard workload, it focuses solely on workload balance and fails to reduce CSTXs. Spring [18] presents a DRL-based address placement approach to reduce CSTXs while balancing the shard workloads. Nevertheless, Spring only addresses the new address placement problem and lacks the ability to adjust in real-time based on the temporary characteristics of transactions, limiting its effectiveness in improving overall system performance. There is still a lack of effective account migration algorithms that both reduce CSTXs and balance shard workloads.

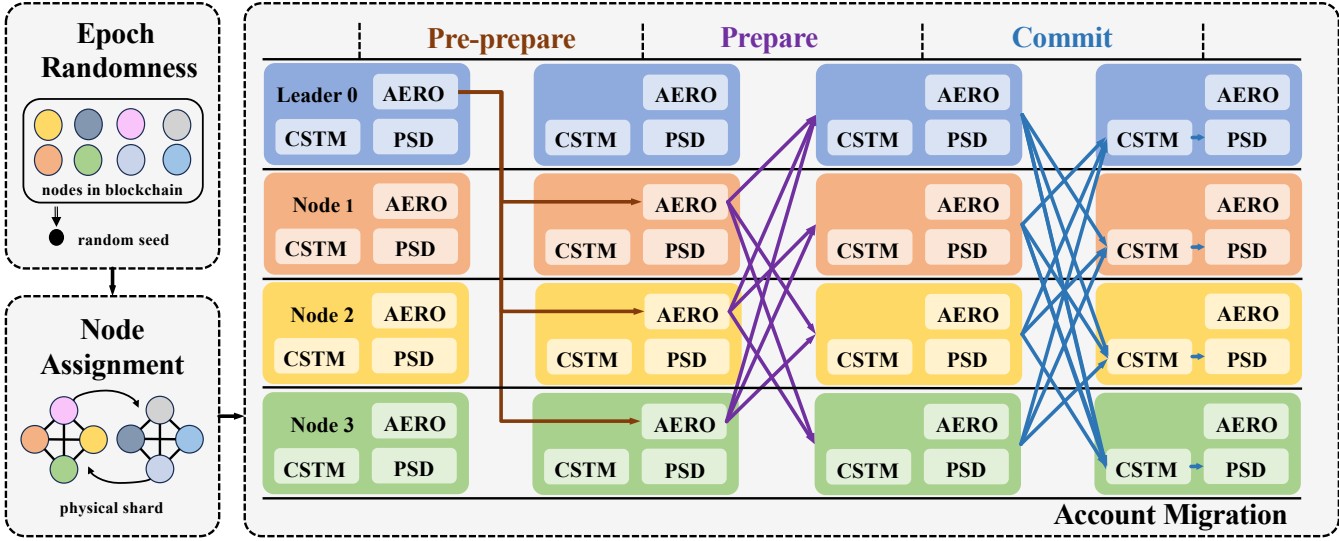

**Figure 1: The reconfiguration phase workflow in AEROChain. After epoch randomness and node assignment, AEROChain goes into account migration. The account migration process adheres to PBFT, with the AERO handling proposal creation and validation. Nodes eventually update their respective Physical Shard Data (PSD) by the Cross-shard Transaction Module (CSTM).**

## 3 AEROChain Design

To validate the feasibility of integrating AERO into a sharding blockchain system, we have developed a prototype system called AEROChain. The design of AEROChain is in the following sections, detailing the structure and functionality of its various components.

### 3.1 Basic System Design

The AEROChain operates on an account-based transaction model. Following the mainstream sharding blockchain design [12, 18, 33], AEROChain assumes a partially asynchronous network [6], where message delays are unbounded but eventual delivery is guaranteed. This assumption aligns with real-world network conditions where latency can vary, but messages are delivered finally. AEROChain is designed with Byzantine Fault Tolerance (BFT), allowing it to tolerate up to $f$ faulty or malicious nodes in a network of $3f + 1$ nodes. AEROChain also resilient an adaptive adversarial model, where an adversary can corrupt nodes dynamically during protocol execution. However, the adaptive adversary is assumed unable to forge or tamper with the signatures of honest nodes, ensuring the integrity of authenticated communications within the network.

### 3.2 Architecture of AEROChain

AEROChain introduces a novel sharding architecture consisting of two types of shards: the *physical shard* and the *logical shard*. Both types utilize the PBFT protocol [3] for achieving consensus. These shards are detailed in the following sections, and the specific components of AEROChain are further explained in Section 3.2.3.

*3.2.1 Physical Shard.* The physical shard is tasked with transaction processing and ledger maintenance. To enhance the parallelism and scalability of AEROChain, the network is partitioned into multiple physical shards, each comprising a subset of nodes. Every node is assigned to one physical shard, where it participates in transaction validation and block creation using the consensus mechanism.

*3.2.2 Logical Shard.* The logical shard is essential for facilitating the migration of account states across physical shards. The logical shard encompasses all nodes in the network, ensuring a low CSTX ratio and a balanced workload between shards. Specifically, it is responsible for generating and executing migration transactions, which are essential for transferring account states between physical shards, thereby supporting efficient load balancing and maintaining the overall system's performance and scalability.

*3.2.3 Components.*

- **Epoch Randomness**: The same random seed is used in each node, and consensus is reached on the same initial trained model parameters. Since subsequent transactions and the state are deterministic, the AERO model updates that follow are also deterministic. This ensures that the nodes in each logical shard can validate the results of the generated migration transactions, guaranteeing consistency across the AEROChain. Moreover, the safety and liveness analysis can be found in Appendix A.
- **Node Assignment**: The random seed is used to periodically reassign and maintain shard nodes, ensuring system security against adaptive adversaries.
- **Physical Shard Data(PSD)**: PSD refers to the information managed and processed within each physical shard. Each physical shard is responsible for transaction validation, block creation, and ledger maintenance. The PSD includes details such as validated transactions, block records, and the current state of accounts within the shard.
- **Cross-shard Transaction Module (CSTM)**: CSTM is to process CSTXs. It employs a relay-based approach based

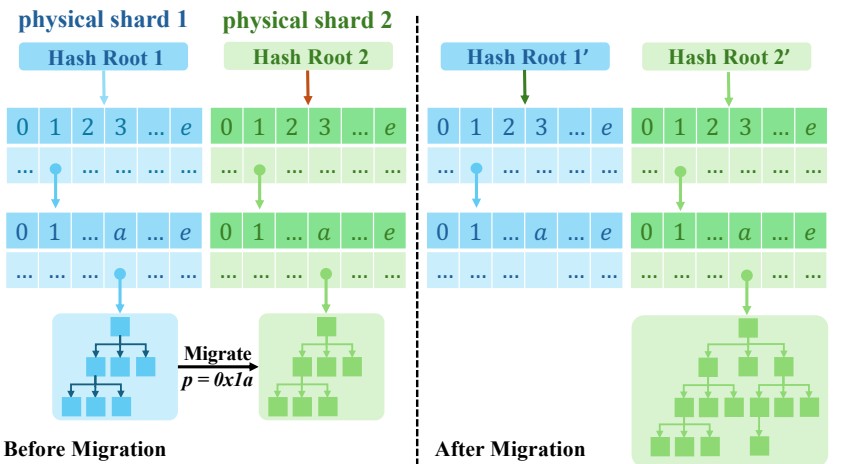

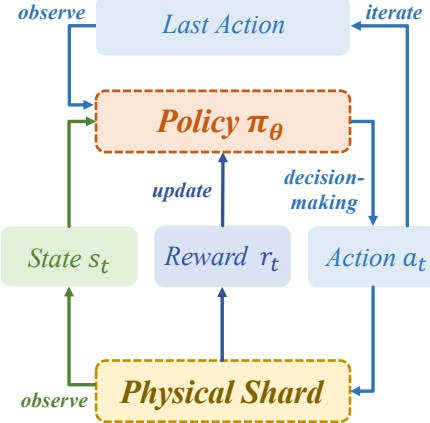

**Figure 2: The execution of a migration transaction {1,2,"0x1a"}.**

**Figure 3: The AERO workflow.**

on the algorithm proposed in Monoxide [29], to coordinate between source and target shards.

- **AERO**: AERO is a DRL-based framework designed to output account migration plans. AERO introduces a prefix-based granularity approach to migrate accounts with the same prefixes, reducing the action space and facilitating efficient migration decisions at group level, rather than for individual accounts. The AERO is described in Section 4.

## 3.3 Workflow of AEROChain

AEROChain operates in time intervals known as epochs, where each epoch is structured into two key phases: the reconfiguration phase and multiple consensus phases.

*3.3.1 Reconfiguration Phase.* At the beginning of each epoch, an epoch randomness is generated to produce a random seed. This seed is then used in the node assignment process to redistribute nodes among physical shards, as well as in the account migration process to ensure a consistent result. By utilizing this randomness in both node assignment and account migration, AEROChain enhances security against adaptive adversaries, making it more difficult for them to predict or target specific shards. Following this, the system proceeds into the Account Migration process.

As shown in Figure 1, the account migration process strictly follows the PBFT protocol during the reconfiguration phase. In the pre-prepare phase, the logical shard leader uses AERO to create a proposal with a migration transaction list $\mathbb{M}$, which contains a series of migration transactions. The leader then broadcasts it to all other nodes in the logical shard. Upon receiving the proposal, each node leverages its AERO to validate the correctness and integrity of the proposal, ensuring the legitimacy of migration transactions.

Once the consensus node has validated the proposal using the AERO, it votes by broadcasting a prepared message to all other nodes during the prepare phase. Subsequently, each node collects prepared messages from others, and when a node has successfully received $2f + 1$ prepare votes, it considers that the proposal has garnered enough votes to proceed to the next phase. This prepare

stage ensures that at least $f + 1$ honest nodes are synchronized and have reached an agreement on the proposal.

In the commit phase, after collecting the required $2f + 1$ commit messages, nodes finalize the agreement and proceed to execute the transactions from $\mathbb{M}$ in proposal. Each node processes the migration transactions related to its physical shard through CSTM. After the execution of the relevant transactions, the node updates the PSD, completing the consensus process.

*3.3.2 Consensus Phase.* Physical shards operate in parallel during the consensus phase, efficiently processing transactions. During this phase, nodes gather and analyze transaction data from each physical shard, summarizing the transaction details between shards in preparation for the account migration process. Transactions processed during this phase fall into two categories: intra-shard transactions and CSTXs, where:

- **Intra-shard Transactions**: Intra-shard transactions involve only sender and receiver accounts within the same shard, allowing them to be processed entirely within that shard without external communication. These transactions benefit from lower latency, as they do not need shard coordination.
- **CSTXs**: CSTXs involving parties from different physical shards. Due to cross-shard communication, CSTXs incur higher overhead compared to intra-shard transactions.

## 3.4 Migration Transactions

Migration transactions are a crucial part of the reconfiguration mechanism, ensuring the secure and efficient migration of account states between physical shards in AEROChain. These specialized CSTXs are triggered by the logical shard to facilitate the transfer of account states from source physical shards to target physical shards. Each migration transaction includes a field $p$, corresponding to the account prefix of the migrating accounts. The migration transaction structure is as shown below:

$$MigrationTransaction = \{sender, receiver, p\}, \qquad (1)$$

where the *sender* and *receiver* fields represent the source and target shards index of the migration account with the prefix $p$. As shown

in Figure 2, the migration transaction is executed by the prefix $p$, from the physical shard sender to the physical shard receiver. In the example provided in the figure, accounts have the prefix $p = 0x1a$ migrates from physical shard 1 to physical shard 2. Initially, both physical shards maintain separate hash roots representing their respective data partitions. After the migration, the account with prefix $0x1a$ resides in physical shard 1. With the account prefixed with $0x1a$ merging from physical shard 1 into physical shard 2, there is no risk of conflict during the migration process. This is because no identical public chain account exists in the shard chain with the same address. Moreover, AERO enables the simultaneous initiation of multiple migration transactions during the reconfiguration phase. Consequently, the migration is guaranteed to be conflict-free. The process of migrating account states involves several steps:

(1) Nodes in the logical shard execute the PBFT to agree on the migration transaction list generated by the leader node.

(2) Once consensus is achieved, the migration transactions are executed in CSTM. Nodes involved in both the source and target shards relay the account states from the source shard to the target shard.

(3) The target shard integrates the migrated account states. After all physical shards have completed this process, AEROChain transitions to the consensus phase, during which the nodes maintaining the target shard update their PSD.

## 4 AERO Design

When the account migration process begins, the policy must assess the current state of each account and shard, reviewing them group by group to determine whether migration is necessary and decide the target destination for the migration. Therefore, the account migration is a standard sequential decision-making process, which can be effectively modeled as a Markov Decision Process (MDP). By framing it as an MDP, AERO can capture the transaction temporal characteristics in sharding blockchain and optimize its policy $\pi_\theta$ with the Proximal Policy Optimization [25], where $\theta$ denotes the parameters of the agent in AERO. Figure 3 is the workflow of AERO. The overall objective of AERO is to reassign accounts to new shards to enhance the overall performance of the sharding blockchain. Specifically, the overall objective function $J(\theta)$ is to minimize the CSTX ratio and shard load variances, which is expressed as:

$$J(\theta) = \mathbb{E}_{\pi_\theta} \left[ \sum_{t=1}^{\infty} \gamma^t (w_1 u_t + w_2 v_t) \right], \tag{2}$$

where $\gamma \in [0, 1)$ is a discount factor, $w_1$ and $w_2$ are weighting coefficients, $u_t$ is the average CSTX ratio in epoch t, and $v_t$ is the average shard workload variances in epoch t. To achieve $J(\theta)$, the optimal policy $\pi_\theta$ is found by:

$$\pi_\theta = \arg\max_\theta J(\theta) \tag{3}$$

$$= \arg\max_\theta \mathbb{E}_{\pi_\theta} \left[ \sum_{t=0}^{\infty} \gamma^t (w_1 u_t + w_2 v_t) \right]. \tag{4}$$

By optimizing $\theta$, the agent can derive a policy that maximizes throughput and minimizes latency. In the following sections, we will provide detailed explanations of the state design, the action representation, the reward function, and the transition dynamics.

### 4.1 State Design

To optimize the account migration strategy, it is crucial to fully analyze how transactions are distributed across the physical shards in the network. By incorporating state variables such as the number of CSTXs and variances in the total transaction volume, we can more precisely capture which shards are experiencing higher loads and denser transaction activity. Importantly, the state from the previous epoch must also be considered to better evaluate the temporal characteristics of transactions over time and assess the effectiveness of the previous migration. This allows for more informed account migration decisions, ultimately improving sharding blockchain performance. The state $\mathbf{s} \in \mathbb{R}^{d_s}$ encapsulates the current status of the physical shards, where $d_s$ is the state dimension. $\mathbf{s}$ includes critical features that influence the account migration decision, such as network load metrics and shard statistics. Specifically, the state $s$ in epoch $t$ is as follows:

$$s_t = \{\mathbb{T}_t, \mathbb{C}_t, \mathbb{V}_t, \mathbb{TX}_t^c, \mathbb{TX}_t^i, \mathbb{TX}_{t-1}^c, \mathbb{TX}_{t-1}^i\}, \tag{5}$$

where the list $\mathbb{T}_t$ represents the throughput of physical shards, the list $\mathbb{C}_\approx$ denotes the overall CSTX ratio of physical shards in epoch $t$, and the list $\mathbb{V}_\approx$ is the variance corresponding to these physical shards. The term $\mathbb{TX}_t^c$ refers to the CSTX volumes for each account prefix $p$ within each physical shard, and $\mathbb{TX}_t^i$ represents the intra-shard transaction volumes. Specially, for the initial state $s_0$, it is defined as $s_0 = \{\mathbb{T}_0, \mathbb{C}_0, \mathbb{V}_0, \mathbb{TX}_0^c, \mathbb{TX}_0^i, \{0\}, \{0\}\}$.

### 4.2 Action Representation

Accurately defining actions that represent the possible migration operations is essential for effectively modeling account migration in an RL framework. Instead of migrating individual accounts one by one, we adopt a prefix-based grouping strategy for account migration in batch. This approach allows RL to better regulate shard states from a macro perspective, significantly simplifying the action space. By mapping each prefix to a corresponding dimension, we facilitate the policy network $\pi_\theta$ in selecting the appropriate prefix $p$ for migration. Incorporating variables that specify which account prefixes to move and between which shards enables the model to capture the flexibility and complexity of migration decisions that directly impact network performance. This strategy allows the model to explore different migration paths, optimize shard utilization, and reduce transaction latency.

Each action involves moving accounts with the specific address prefix to new shards, and the action $\mathbf{a}_t \in \mathbb{R}^{n_t \times 3}$ defines migration operations to be performed at epoch $t$. Here, $n_t$ varies depending on the current state and context, allowing for variable-length action sequences. Consequently, the action $\mathbf{a}_t$ is defined as:

$$\mathbf{a}_t = [\mathbf{a}_t^{(1)}, \mathbf{a}_t^{(2)}, \ldots, \mathbf{a}_t^{(i)}, \ldots, \mathbf{a}_t^{(n_t)}], \tag{6}$$

$$\mathbf{a}_t^{(i)} = (A_t^{(i1)}, A_t^{(i2)}, p), \tag{7}$$

where $A_1$ and $A_2$ refers to the source and target shard index.

The neural network is utilized to capture complex temporal transaction dependencies in the sharding blockchain. Moreover, AERO also employs a sliding window mechanism that captures the most recent migration transactions, ensuring that the model focuses on the most relevant information without being overwhelmed by

the entire history. We consider the state $\mathbf{s}$ and the action history list $\mathbf{a}$ as the input.

In each node participating in the logical shard, the same random seed is used, and consensus is reached on the same initial trained model parameters. Since subsequent transactions and the state are deterministic, the AERO model updates that follow are also deterministic. This ensures that the nodes in each logical shard can validate the results of the generated migration transactions, guaranteeing consistency across the sharding blockchain. Moreover, the safety and liveness analysis can be found in Appendix A.

### 4.3 Reward Function

The reward function aims to lead AERO optimizing throughout by balancing CSTX ratio and the shard workload variances. The reward at epoch $t$ is defined as:

$$R_t = w_1 u_t + w_2 v_t, \tag{8}$$

where $w_1$ and $w_2$ are weighting coefficients. The term $u_t$ is the average CSTX ratio in epoch $t$:

$$u_t = \frac{c}{b_t + c_t}, \tag{9}$$

$$c_t = \frac{1}{N} \sum_{i=1}^{N} \text{CST}_t^i, \tag{10}$$

$$b_t = \frac{1}{N} \sum_{i=1}^{N} \text{IST}_t^i, \tag{11}$$

and $v_t$ is the negative variance of CSTX ratio in epoch $t$:

$$v_t = -\sigma^2 = -\frac{1}{N} \sum_{i=1}^{N} (\text{CST}_i - c_t)^2, \tag{12}$$

here, $\text{CST}_i$ is the number of CSTXs in shard $i$, $\text{IST}_i$ is the number of intra-shard transactions in shard $i$, and $N$ is the total number of shards. By maximizing $R_t$, the agent is encouraged to reduce $u_t$ and minimize $v_t$, leading to balanced and efficient system performance.

### 4.4 Transition Dynamics

The state transition function models how the environment responds to the actions and the inherent temporal characteristics of incoming transactions. The next state is influenced by both the current state and upcoming transactions, which cannot be directly measured or predicted. However, transactions exhibit temporal patterns, and the actions taken by the agent can affect these patterns. Therefore, the next state is given by:

$$\mathbf{s}_{t+1} = f(\mathbf{s}_t, \mathbf{a}_t, \mathbf{w}_t), \tag{13}$$

where $\mathbf{w}_t$ represents stochastic factors such as network fluctuations and unobservable upcoming transactions. The function $f$ captures the complex interactions between the current state, the action taken, and the stochastic elements of the sharding blockchain.

Understanding the transition dynamics is challenging due to the unobservable nature of future transactions and their dependency on both temporal patterns and the agent's actions. Despite this, the agent can learn these dynamics through observed state, action, and reward sequences. By capturing the temporal dependencies and learning from the environment, the agent optimizes long-term rewards. It adapts its strategy to the temporal characteristics of

transaction flows and the stochastic nature of the environment, enhancing the overall efficiency and robustness of the system.

## 5 Evaluation

### 5.1 Experimental Settings

AEROChain is developed in Golang and the AERO is implemented in Python, with a total codebase exceeding 4,000 lines. AEROChain's implementation is based on BlockEmulator [11], which provides a scalable sharding blockchain environment. The AERO is built upon the cleanrl framework[13], facilitating the development of RL with a focus on clarity and simplicity. The experimental setup consists of 16 physical shards, each containing 8 nodes, amounting to a total of 128 nodes in the total network. During each epoch, the consensus phase is composed of 100 blocks, with each block containing a maximum of 1,000 transactions.

To ensure our experiments reflect actual network conditions and transaction patterns, we utilized real transaction data from Ethereum in 2024 [35]. We employed 1 million transactions to test performance in a real-world environment, providing insights into AERO's effectiveness in optimizing CSTXs and load balancing compared to other algorithms. Additionally, 10 million transactions are used to train the AERO model, enabling it to learn and adapt to the complex transaction patterns inherent in blockchain. The hyperparameters are detailed in Appendix B.

### 5.2 Baselines

To comprehensively assess the performance of AERO, we have selected five algorithms for comparison:

(1) **AERO-S**. AERO-S is an implementation with individual-by-individual migration of AERO, designed to explore the efficiency of the group migration mechanism on capturing complex transaction patterns.

(2) **Spring** [18]. SPRING uses DRL to optimize state placement in the sharding blockchain, reducing CSTXs and improving the blockchain throughput. We choose SPRING to compare the effectiveness of account migration and account allocation in improving CSTXs and load balance.

(3) **BrokerChain** [10]. BrokerChain uses graph partitioning to optimize state partitioning and account segmentation, with the goal of balancing transaction workloads and minimizing CSTXs. We leverage Broker to evaluate graph partitioning.

(4) **LB-Chain** [17]. LB-Chain uses a load-balancing approach to dynamically balance transaction workloads across shards. We use LB-Chain to compare the effectiveness of strategies focused on optimizing transaction workloads.

(5) **Monoxide** [29]. Monoxide improves transaction processing by using asynchronous consensus zones to handle CSTXs efficiently. We use this algorithm to evaluate AEROChain without introducing any account migration strategies.

### 5.3 Overhead Analysis

Integrating AERO into AEROChain introduces both computational and storage overheads, which we analyze in this section.

**Storage Overheads**. The AERO model in AEROChain occupies approximately 90KB of disk. This compact size ensures that the

storage requirements do not impose significant burdens on the nodes, allowing for efficient deployment across the network.

**Computational Overheads**. The computational overhead of AERO involves two key components: (1) the time spend generating and validating migration transactions, which includes the duration the AERO agent takes to decide which accounts to move between shards, and (2) the time required to update the training model, which occurs once per reconfiguration phase.

In our experiments, migration decisions take approximately 0.06 seconds per decision, while updating the training model required around 0.9 seconds on the hardware used. These times are notably faster than those observed with graph partitioning-based methods, highlighting the efficiency of AERO.

## 5.4 Cross-Shard Transaction Ratio

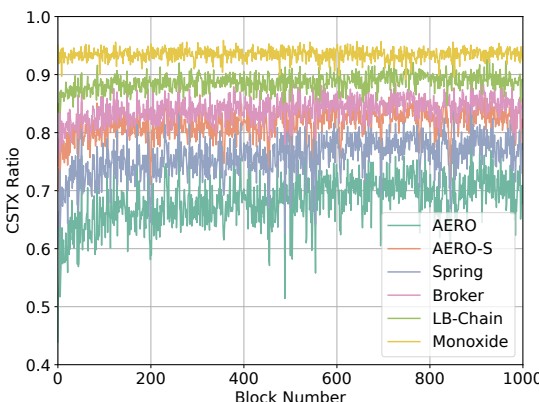

**Figure 4: The CSTX ratio over 1000 block.**

The purpose of this experiment is to evaluate the effectiveness of different migration strategies in reducing CSTX ratio, which is crucial for improving overall blockchain performance and efficiency. As shown in Figure 4, the performance in reducing CSTX ratios is largely influenced by strategies for account management. AERO demonstrates a relatively low CSTX ratio. The DRL architecture enables AERO to effectively capture patterns in account activities, allowing for the dynamic accounts migration to shards where they engage in frequent transactions. By allocating these accounts within the same shard, AERO successfully reduces CSTXs. AERO-S shows low effectiveness, as its simpler migration mechanism struggles to fully capture the intricate and dynamic dependencies between accounts. This limitation diminishes its overall efficiency.

Spring shows a higher CSTX ratio compared to AERO, primarily because its approach focuses more on optimizing state placement rather than account migration. While optimizing state placement can reduce some cross-shard interactions, it lacks the dynamic adaptability to handle the complex and changing relationships between accounts. As a result, Spring's higher ratio suggests that static or less adaptive methods are not as effective in managing CSTXs in a highly dynamic blockchain environment. Broker and LB-Chain are not very effective and have quite high CSTX ratios, which can be attributed to their focus on balancing workloads across shards. Broker uses graph partitioning techniques to optimize the state partitioning, which may still have limitations when handling dynamic account migration issues. LB-Chain effectively prevents shard overload but struggles with reducing CSTX. Monoxide is

not able to reduce CSTX as effectively as other algorithms, which highlights the importance of efficient transaction processing and a well-designed account management strategy.

## 5.5 Shard Load Variance

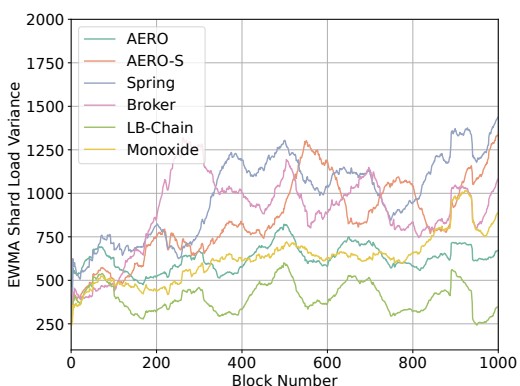

**(a) The EWMA shard load variance.**

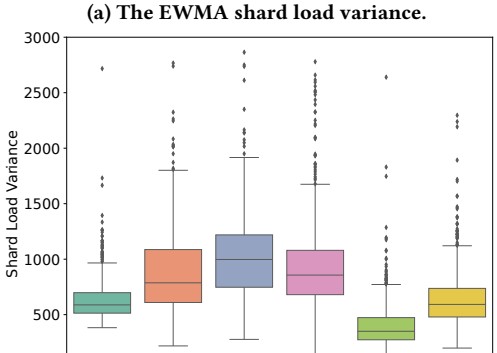

**(b) Box plot of shard load variance.**

**Figure 6: Shard load variance for different algorithms.**

This experiment aims to evaluate the effectiveness of different algorithms in balancing shard workloads, as measured by shard load variance. Specifically, we investigate how timely migrating accounts from heavily loaded shards to less loaded ones can reduce load imbalances. Figure 6a presents the exponentially weighted moving average (EWMA) variance of shard load balance. EWMA is adopted to smooth the data and highlight trends in load variation over time. As illustrated in Figure 6, the shard load variance highlights distinct patterns driven by the underlying mechanisms for balancing shard workloads. AERO maintains a low load variance by dynamically migrating accounts based on interaction patterns, ensuring a more balanced distribution of shard workloads. By effectively managing these migrations, AERO reduces imbalances and keeps the variance in shard load relatively low compared to other algorithms. AERO-S shows higher variance than AERO due to its simpler mechanism, which limits its ability to balance workload across shards.

Spring exhibits a high shard load variance, stemming from its strategy of optimizing state placement. Although this approach can mitigate some types of imbalances, it lacks the flexibility to adapt to shifting transaction patterns, leading to a more uneven distribution of transactions across shards over time. Broker performs slightly

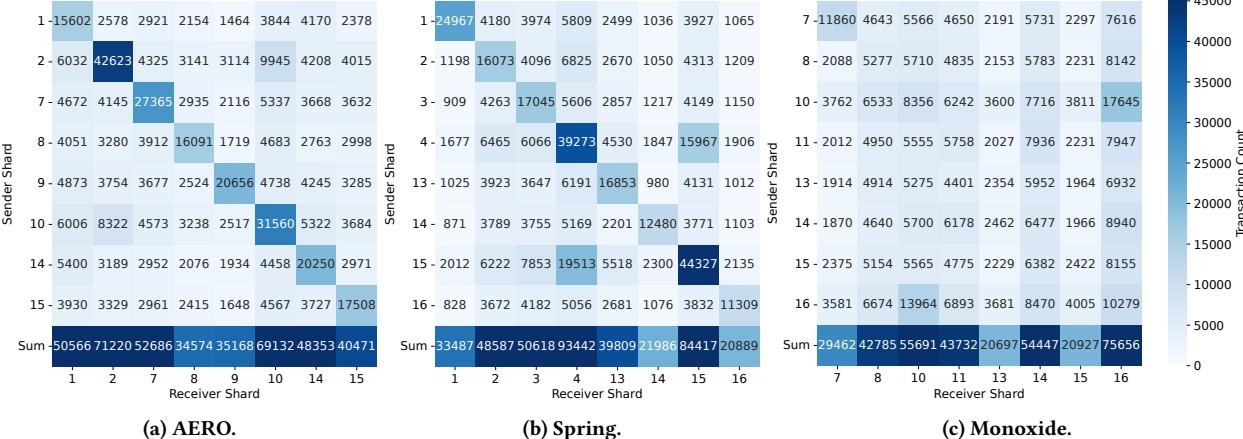

(a) AERO.           (b) Spring.           (c) Monoxide.

**Figure 5: The comparison of transaction distribution for different algorithms during 1,000 block numbers.**

better than Spring as it addresses migration issues rather than account allocation. However, it remains focused on reducing CSTX, which limits its ability to dynamically adapt to transaction behavior changes, resulting in only moderate improvements in shard load variance. LB-Chain demonstrates stronger performance in load balancing, with its mechanism effectively preventing significant overloads. Monoxide has a strong performance in minimizing shard load variance, primarily because the absence of account migration results in a more even distribution of accounts across shards.

## 5.6 Shard Transaction Distribution

The purpose of this experiment is to investigate how the internal distribution of shard workload affects the external performance of different blockchain protocols. This can also provide insights into the protocols' overall efficiency and ability to manage imbalances. The heatmaps in Figure 5 illustrate the transaction distribution across shards for three representative algorithms over 1,000 blocks. The figure highlights eight shards with distinct transaction patterns, while the complete shard distribution can be found in Appendix D. Starting with AERO, the heatmap reveals a strong concentration of intra-shard transactions, as indicated by the dark diagonal line that runs from the top left to the bottom right of the chart. This suggests that AERO is highly effective in grouping frequently interacting accounts within the same shard, thereby significantly reducing the need for CSTXs. The lighter shades in the off-diagonal regions shows that AERO minimizes cross-shard interactions.

In contrast, Spring shows a less concentrated distribution of intra-shard transactions, with more noticeable imbalances across shards. Certain shards exhibit significantly higher CSTX volumes, and this uneven distribution of CSTXs implies that Spring struggles to maintain a balanced workload distribution and reduces overall system performance due to increased CSTX load. Monoxide exhibits the weakest performance among the three algorithms, though it shows a more balanced distribution of cross-shard transactions. The heatmap reflects a relatively uniform spread of transactions across both diagonal and off-diagonal regions, which means that the algorithm generates the most CSTXs, negatively impacting system throughput. Although the transaction distribution is more balanced compared to Spring, the overall CSTX burden is significantly higher, leading to inefficiencies in handling workloads.

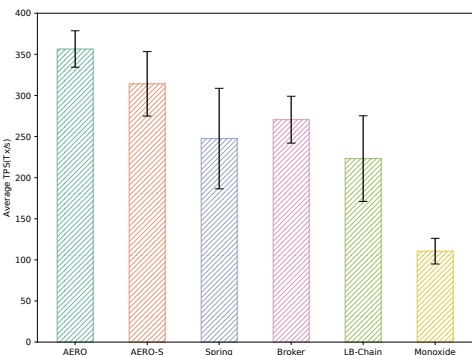

**Figure 7: Comparison of overall TPS.**

## 5.7 Overall Throughput Analysis

Finally, we assessed the overall throughput of AEROChain in terms of TPS. Figure 7, which compares the average TPS across different algorithms, highlights AERO's performance over the state-of-the-art algorithm by 31.77%. When compared to AERO-S, AERO shows a clear improvement, indicating that group migration mechanism plays a crucial role in enhancing throughput. Spring and Broker perform less efficiently than AERO, primarily due to their reliance on less adaptive strategies that struggle with fluctuating transaction patterns. Their higher CSTX ratios and uneven load balance further reduce the overall throughput. Monoxide, which has the lowest TPS, is constrained by its lack of account migration strategies. While its asynchronous consensus zones allow for efficient CSTX processing, Monoxide struggles at reducing the CSTX number, leading to lower throughput in sharding blockchain.

## 6 Conclusion

We proposed AERO, an attention-based DRL framework designed for efficient account migration in sharding blockchain networks. By employing a prefix-based granularity approach to reduce the action space and integrating attention mechanisms to capture temporal characteristics, AERO effectively minimizes CSTXs and balances workload across shards. Our implementation of sharding blockchain system, AEROChain, demonstrates the feasibility and adherence to blockchain decentralization principles of the AERO. Extensive evaluation with real Ethereum data show that AERO outperforms existing solutions, improving system throughput by 31.77%.

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

# A  Safety and Liveness of AERO

The account migration process emphasizes decentralization and security, leveraging the byzantine fault tolerance guaranteed by the PBFT protocol. Additionally, in the case of a leader failure or other disruptions, the view-change phase is handled in accordance with PBFT's specifications, ensuring that a new leader is selected and the process can continue without compromising the safety and liveness of the network. Through the cooperation of various components, AEROChain realizes a decentralized and secure account migration process, ensuring consistency and integrity across the network. This section analyses the safety and liveness properties specific to generating and executing the account migration plan.

LEMMA 1. *Assuming that malicious nodes constitute less than one-third of all consensus nodes within the logical shard, the account migration plan can ensure safety.*

PROOF. In AERO, all consensus nodes are required to serve as consensus nodes for both the logical and physical shards. We assume that malicious nodes constitute less than one-third of the nodes in each shard. Consequently, the number of malicious nodes within the logical shard remains under one-third of the total.

The account migration plan is generated through a consensus phase, ensuring that the honest nodes can accept no malicious or flawed migration plan. Therefore, AERO can guarantee safety if fewer than one-third of the consensus nodes are malicious.  □

LEMMA 2. *Assuming that malicious nodes constitute less than one-third of all consensus nodes within the logical shard, the account migration plan can ensure liveness.*

PROOF. In the AERO design, we operate under the assumption of a partially synchronous network model, where message delays are bounded by an unknown time parameter, denoted as $\delta$. This model implies that while network delays may be unpredictable, all messages are guaranteed to be delivered within a finite, uncertain time frame. Under these conditions, the migration plan generated by the logical shard will eventually be transmitted to the target shard. Once received, the account migration process will be completed, ensuring both liveness and eventual consistency in the system. □

## B Hyper-parameters Settings

**Table 1: Hyperparameters and their values**

| Hyperparameter | Value |
|---|---|
| Discount factor, $\gamma$ | 0.99 |
| Numbers of heads in transformer, $h$ | 6 |
| Batch size | 128 |
| Mini batch size | 4 |
| Learning rate | 1e-5 |
| Number of neurons in each layer | 256 |

## C Neural Network Design

Incorporating attention mechanisms into DRL models can help the agent gain a deeper understanding of the global state of the shards, enabling it to capture transaction patterns and relationships between accounts [4, 7]. AERO generates the queries, keys, and values by utilizing the $\mathbf{a}$ and $\mathbf{s}$. Specifically, the query vector $\mathbf{Q}$ is derived from $\mathbf{s}$, while the key and value matrices, $\mathbf{K}$ and $\mathbf{V}$, are obtained from the encoded action history $\mathbf{H}_{\text{enc}}$. It is represented as follows:

$$\mathbf{Q}_i = \mathbf{W}_{q_i}\mathbf{a},$$
$$\mathbf{K}_i = \mathbf{W}_{k_i}\mathbf{s}, \qquad (14)$$
$$\mathbf{V}_i = \mathbf{W}_{v_i}\mathbf{s},$$

where $\mathbf{W}_{q_i} \in \mathbb{R}^{d_h \times d_s}$ is the query projection matrix, $\mathbf{W}_{k_i}, \mathbf{W}_{v_i} \in \mathbb{R}^{d_h \times d_{\text{model}}}$ are the key and value projection matrices, and $\mathbf{H}_{\text{enc}} \in \mathbb{R}^{L \times d_{\text{model}}}$ is the encoded representation of the action history, with $L$ being the sequence length after encoding. Here, $d_h$ denotes the dimension of the hidden layer in the attention mechanism. By using these attention weights to the value vectors, AERO aggregates from the action history and state to form a context vector $\mathbf{h}$:

$$\mathbf{h}_i = \text{softmax}\left(\frac{\mathbf{Q}_i\mathbf{K}_i^\top}{\sqrt{d_h}}\right)\mathbf{V}_i, \qquad (15)$$

$$\mathbf{h} = [\mathbf{h}_1, \mathbf{h}_2, \ldots, \mathbf{h}_h]\,\mathbf{W}_O, \qquad (16)$$

where $\mathbf{W}_O \in \mathbb{R}^{h \cdot d_h \times d_{\text{model}}}$ is the output projection matrix that combines the outputs from all attention heads into a single context vector $\mathbf{h}$. $\mathbf{h}$ encapsulates the most pertinent information needed to make informed migration decisions. The decoder then generates variable-length action sequences in an autoregressive manner. At each decoding step $k$, the model produces an action using:

$$\mathbf{a}_t^{(k)} = \mathbf{W}_{\text{out}}\mathbf{h}^{(k)} + \mathbf{b}_{\text{out}}, \qquad (17)$$

where $\mathbf{W}_{\text{out}} \in \mathbb{R}^{3 \times d_{\text{model}}}$ is the action projection matrix, $\mathbf{b}_{\text{out}} \in \mathbb{R}^3$ is the bias term, $\mathbf{h}^{(k)}$ is the context vector $\mathbf{h}$ at step $k$, and $\mathbf{a}_t^{(k)} \in \mathbb{R}^3$ is the predicted action at step $k$. The decoder continues to generate actions until an end-of-sequence token is produced or a maximum sequence length is reached.

## D Full Shard Transaction Distriction

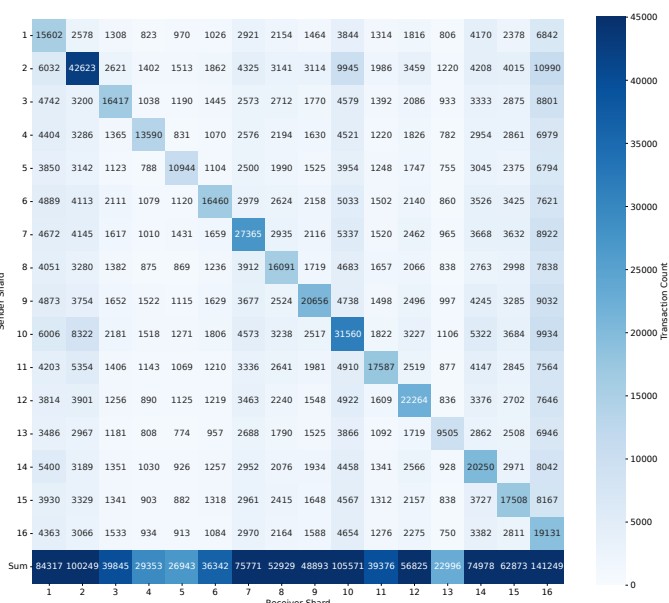

**Figure 8: The transaction distribution of AERO.**

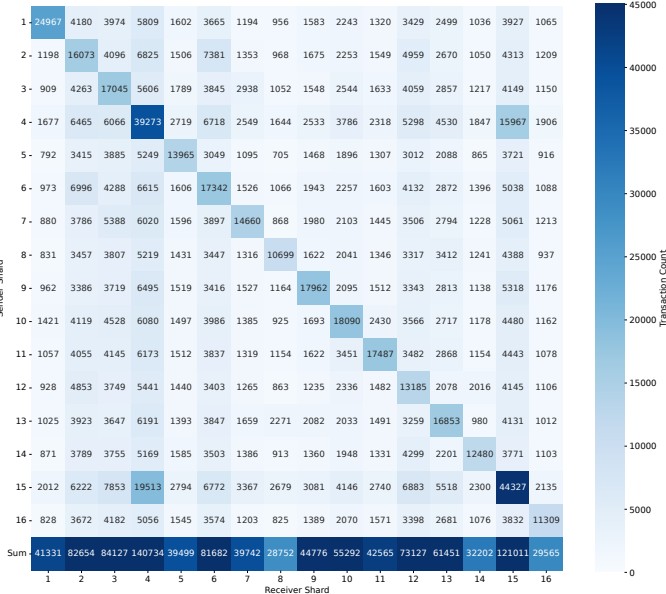

**Figure 9: The transaction distribution of Spring.**

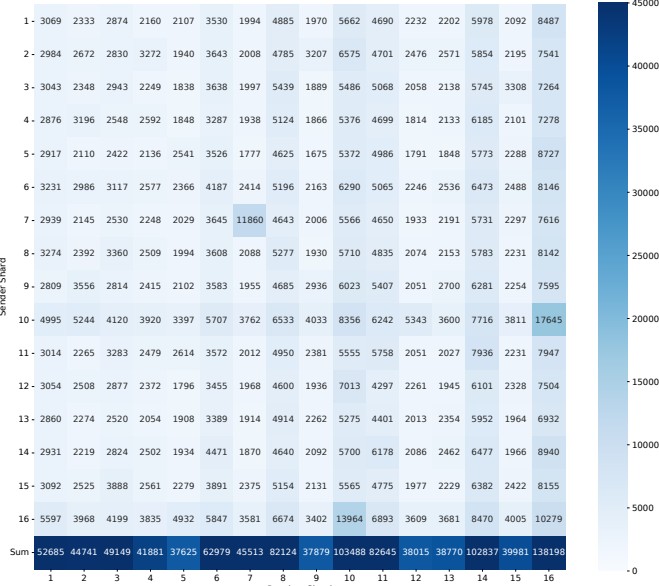

**Figure 10: The transaction distribution of Monoxide.**