# OpenReview forum: "AERO: Enhancing Sharding Blockchain via Deep Reinforcement Learning for Account Migration"
_ACM.org/TheWebConf/2025/Conference — WWW 2025 Oral_

### Official Review · Reviewer_vQLF · 2024-11-02

**Novelty:** 4
**Technical Quality:** 5

**Review:**

### Summary:
The paper introduces AERO, a deep reinforcement learning-based approach for effectively migrating accounts across blockchain shards.
Compared to other approaches, it focuses on account migration to reduce cross-shard transactions and not, for example, on balancing the transaction load per shard.
The paper evaluates AERO using a dedicated sharding blockchain system called AEROChain and further compares it to other state-of-the-art approaches (including Spring, Broker, LB-Chain, and Monoxide).
The evaluation, which utilizes real Ethereum transaction data, indicates that the account migration approach outperforms related work in terms of reducing the fraction of cross-shard transactions.

### Pros:

+1: The proposed design, with its new focus, outperforms related work.

+2: The evaluation focuses on several important influences (including overhead).

+3: The evaluation sources real Ethereum transaction data.

### Cons:

-1: The paper structure is odd since AEROChain is being introduced before AERO is detailed.

-2: A discussion of how AERO can be integrated into other blockchain designs is missing.

Thank you for submitting this work; it has been an interesting read.
The proposed approach appears to be a good candidate for reducing cross-shard transactions.
I like the elaborate evaluation (as well as the additional details in the appendix), which gives a good intuition about the proposal's performance/impact.

As I detail below, from my point of view, a few minor issues impair the paper's quality, but nothing too critical.
I hope that my comments allow you to further improve your presentation.

### Detailed Comments:

#### -1: Paper Organization

Personally, I think that the current paper structure is a little bit odd because the paper first introduces an auxiliary part (i.e., AEROChain) before detailing the proposed approach, AERO, shifting the focus to the wrong thing.
Moreover, this approach also raises the question of AERO's compatibility with other blockchain designs since AEROChain is very prominent and seems to be fundamental for the remainder of the paper (see also -2).
Considering rearranging the paper's structure could be a good idea to emphasize AERO.

#### -2: Generalizability

The paper currently evaluates AERO using its own blockchain design, AEROChain, raising the question of how AERO translates to other blockchains that feature sharding.
Consequently, adding a discussion on this matter would probably improve the paper and provide relevant (background) information when assessing the proposed concept.

#### Other:
 - How is the prefix length chosen/configured? What kind of implications does it have? The example just talks about a prefix of length 2. How does it scale/translate to other settings?
 - Section 3.3: I feel that there is an issue because the paper talks once about "2f + 1" and once about "f + 1". Should the second occurrence also refer to "2f + 1"?
 - A paragraph that outlines the differences between a physical and logical shard could be added to the paper (if needed in the appendix) to ease novice readers' understanding of this concept.
 - The paper appears to evaluate AERO just using a single chunk of transactions. Have any measures been taken to verify the statistical significance of the reported numbers (compare cross-validation)?
 - Can you elaborate on the ideal trade-off between account migration, workload balancing, and preventing a shard overload? Is there a general answer to this question? Which influences should be considered?
 - Are you committed to open science? Will you publicly share/open-source the research artifacts?

### Nits:
 - Several spaces are missing, e.g., in front of citation markers or opening brackets. Please fix these issues.
 - The writing style of "SPRING" is inconsistent throughout the paper. I recommend always using the capitalized version
 - Introduction: "despite its revolutionary" -- I think that the wording/word form is incorrect; for example, add "impact" or "character".
 - Figure 2: "physical shard" could/should be capitalized, in my opinion.

**Questions:**

- How is the prefix length chosen/configured? What kind of implications does it have? The example just talks about a prefix of length 2. How does it scale/translate to other settings?
 - Section 3.3: I feel that there is an issue because the paper talks once about "2f + 1" and once about "f + 1". Should the second occurrence also refer to "2f + 1"?
 - The paper appears to evaluate AERO just using a single chunk of transactions. Have any measures been taken to verify the statistical significance of the reported numbers (compare cross-validation)?
 - Are you committed to open science? Will you publicly share/open-source the research artifacts?

**Reviewer Confidence:**

3: The reviewer is confident but not certain that the evaluation is correct

**Scope:**

3: The work is somewhat relevant to the Web and to the track, and is of narrow interest to a sub-community

---

### Official Review · Reviewer_nQDL · 2024-11-28

**Novelty:** 5
**Technical Quality:** 5

**Review:**

This work aims to achieve efficient account migration with fairness in the sharding blockchain system. The strengths of this work primarily derive from two aspects. First, in the design of the reinforcement learning (RL) algorithm, group-based action abstraction is employed to enable efficient transaction migration and historical features are stacked in the feature engineering process to account for state dynamics. Second, the work implements a transaction migration system based on PBFT protocol, which integrates an RL-based transaction migration module into the system. The problem is well-defined, and the experiments support the proposed approach. However, several design aspects require further explanation.

**Questions:**

- In Equation (2), the index of the summation symbol should start from 0.
- Although group-based actions are used to improve efficiency, the size of the action space still grows exponentially with the number of migratable shards and transaction groups to be migrated. Given this background, how does the algorithm ensure effective training and convergence?
- Load balancing and the quantity of cross-partition transactions appear to be trade-off objectives. Supplementing the experiments to explore the trade-off between these two factors is recommended.

**Reviewer Confidence:**

4: The reviewer is certain that the evaluation is correct and very familiar with the relevant literature

**Scope:**

4: The work is relevant to the Web and to the track, and is of broad interest to the community

---

### Official Review · Reviewer_3bMu · 2024-11-29

**Novelty:** 5
**Technical Quality:** 6

**Review:**

The paper proposes AERO, which aims to reduce cross-shard transactions (CSTX) and balance shard workload in sharded blockchains. AERO employs deep reinforcement learning (DRL) to decide account states. AERO is integrated into a blockchain system, AEROChain, which uses reconfiguration and consensus phases to redistribute accounts among shards and process transactions. The system is evaluated using historical Ethereum data, demonstrating significant improvements in throughput and CSTX reduction compared to existing solutions.


# Pros
- A novel group-level account migration approach to improve throughput.
- The evaluation is detailed and considers multiple baselines.
- The paper is well-structured and well-written.

# Cons
- The system may not be scalable with real-world blockchain sizes and transaction volumes.
- Given the existence of SPRING, the novelty of employing DRL needs to be justified.

**Questions:**

- Section 5.7. The experiment is still conducted over 1,000 blocks, correct? 1,000 blocks is quite small compared to real-world blockchains. Will the improvement in throughput be scalable with your solution under a larger blockchain?

- Will your artifact and code be made publicly available?

**Reviewer Confidence:**

3: The reviewer is confident but not certain that the evaluation is correct

**Scope:**

3: The work is somewhat relevant to the Web and to the track, and is of narrow interest to a sub-community

---

### Official Review · Reviewer_c4Zs · 2024-12-02

**Novelty:** 4
**Technical Quality:** 5

**Review:**

This paper presents AERO, a deep reinforcement learning framework designed to optimize account migration in sharding blockchains. AERO employs a prefix-based grouping strategy to capture transaction patterns and account relationships. Integrated within AEROChain, it adheres to decentralization principles. Evaluations using real Ethereum data demonstrate that AERO improves throughput by 31.77%, reduces cross-shard transactions, and balances shard workloads more efficiently than existing solutions.

##Pros
- Well-written and well-organized, making it easy to follow.
- Clear design.
- Promising performance results.

##Cons
- The method for determining the prefix and the impact of prefix length are unclear.
- The settings for the five comparison algorithms in Section 5.2 are not explained.
- Some terminology lacks consistency, such as the use of both "SPRING" and "Spring" in the paper.

**Questions:**

1. How is the prefix determined?
2. Does the length of the prefix impact performance?

**Reviewer Confidence:**

2: The reviewer is willing to defend the evaluation, but it is likely that the reviewer did not understand parts of the paper

**Scope:**

3: The work is somewhat relevant to the Web and to the track, and is of narrow interest to a sub-community